# Impact of Lowering TSH Cut-Off on Neonatal Screening for Congenital Hypothyroidism in Minas Gerais, Brazil

**DOI:** 10.3390/ijns10030052

**Published:** 2024-07-18

**Authors:** Nathalia Teixeira Palla Braga, Jáderson Mateus Vilela Antunes, Enrico Antônio Colosimo, Vera Maria Alves Dias, José Nélio Januário, Ivani Novato Silva

**Affiliations:** 1Pediatric Endocrinology Service, Hospital das Clínicas, Universidade Federal de Minas Gerais, Av. Alfredo Balena 190, Belo Horizonte 30130-100, Brazil; jad_antunes@yahoo.com.br (J.M.V.A.); verwaldias@gmail.com (V.M.A.D.); 2Department of Statistics, Universidade Federal de Minas Gerais, Av. Presidente Antônio Carlos 6627, Belo Horizonte 31270-901, Brazil; enricoc57@gmail.com; 3Center for Actions and Research in Diagnostic Support (NUPAD in Portuguese), Medicine Internal Department/Medical School, Universidade Federal de Minas Gerais, Av. Alfredo Balena 190, Belo Horizonte 30130-100, Brazil; nelio@nupad.medicina.ufmg.br; 4Pediatric Endocrinology Service, Hospital das Clínicas, Pediatrics Department/Medical School, Universidade Federal de Minas Gerais, Av. Alfredo Balena 190, Belo Horizonte 30130-100, Brazil; ivanins@gmail.com

**Keywords:** congenital hypothyroidism, neonatal screening, cut-off, sensitivity

## Abstract

A higher incidence of primary congenital hypothyroidism (CH) has been related to increased sensitivity in neonatal screening tests. The benefit of treatment in mild cases remains a topic of debate. We evaluated the impact of reducing the blood-spot TSH cut-off (b-TSH) from 10 (Group 2) to 6 mIU/L (Group 1) in a public neonatal screening program. During the study period, 40% of 123 newborns with CH (*n* = 162,729; incidence = 1:1323) had b-TSH between 6 and 10 mIU/L. Group 1 patients had fewer clinical signs (*p* = 0.02), lower serum TSH (*p* < 0.01), and higher free T4 (*p* < 0.01) compared to those in Group 2 at diagnosis. Reducing the b-TSH cut-off from 10 to 6 mIU/L increased screening sensitivity, allowing a third of diagnoses, mainly mild cases, not being missed. However, when evaluating the performances of b-TSH cut-offs (6, 7, 8, 9, and 10 mIU/L), the lower values were associated with low positive predictive values (PPVs) and unacceptable increased recall rates (0.57%) for a public health care program. A proposed strategy is to adopt a higher b-TSH cut-off in the first sample and a lower one in the subsequent samples from the same child, which yields a greater number of diagnoses with an acceptable PPV.

## 1. Introduction

Neonatal screening for congenital hypothyroidism (CH) is a milestone in the prevention of severe neurological sequelae, enabling the timely identification and treatment of affected newborns [1,2,3]. Immediately after screening was implemented, the incidence of the disease increased from 1:6700 to approximately 1:3500 live births. In recent decades, however, higher incidences have been reported, between 1:1400 and 1:3000, depending on the region, ethnicity, and screening methods [1,2,4,5,6,7].

The reason for this increase is still debated. Longer survival of premature and low-birthweight newborns, advanced maternal age, and ethnic changes in the population have been suggested as contributing factors [7,8,9,10]. Nonetheless, a decreased thyroid-stimulating hormone (TSH) cut-off at blood-spot screening (b-TSH) is considered the main factor for the increased incidence [2,11,12,13]. This rise occurs primarily, but not exclusively, because of the detection of mild cases with eutopic glands [4].

Treatment with levothyroxine for moderate to severe CH has unquestionable benefits. However, the natural course of the disease and the need for treatment in mild cases are not fully understood [2,14,15,16,17]. This is one of the reasons why it is necessary to discuss the need and impact of decreasing the b-TSH cut-off level in neonatal screening programs for CH.

With this in mind, we evaluated the impact of b-TSH cut-off reduction on a government-funded neonatal screening program for CH.

## 2. Materials and Methods

This universal prospective cohort study was conducted between November 2021 and August 2022 in the Neonatal Screening Program funded by the Brazilian State of Minas Gerais (PTN-MG, acronym in Portuguese).

Minas Gerais is the 4th largest (588,383 km^2^) and 2nd most populated Brazilian state, with 853 municipalities. It is located in the south-eastern region of Brazil. Approximately 18,000–20,000 newborns are screened per month by the program.

The study was approved by the Research Ethics Committee of Universidade Federal de Minas Gerais (CAAE 50311321.1.0000.5149), the parents or legal guardians received relevant information and signed an informed consent form.

Before the study, the PTN-MG used the protocol established by the Brazilian Ministry of Health to measure filter-paper bloodspot TSH (b-TSH) with a cut-off of 10 mIU/L. During the study, the protocol for newborn screening was modified, and the cut-off value for b-TSH was reduced from 10 mIU/L to 6 mIU/L.

The technical operation of the PTN-MG is carried out by the Núcleo de Ações e Pesquisa em Apoio Diagnóstico (NUPAD) [Center for Diagnostic Support Action and Research] of the Medical School of the Universidade Federal de Minas Gerais (FM-UFMG). General management and coordination are performed by the Health Secretariat of the State of Minas Gerais.

Bloodspots on filter paper (S&S 903^®^) are collected, between the third and fifth day of life, at primary healthcare units (90%) or birth hospitals (10%), from newborns who were not discharged by the fifth day due to relevant clinical conditions. There are 3744 collection points. Samples are sent via the postal service to the referenced laboratory at Nupad, and results are typically released within 24 h of sample receipt. Given the vast size of Minas Gerais, the average transportation time from sampling to arrival at the laboratory may vary, up to 5 days.

b-TSH is evaluated using the fluoroimmunoassay method (GSP Neonatal hTSH, Turku, Finland) at the NUPAD laboratory. According to the protocol, children with b-TSH ≥ 20 mIU/L were contacted by phone for an emergency appointment at the reference center to confirm the diagnosis. Those who presented borderline results (between 6 and 20 mIU/L) had a second sample immediately collected on a filter paper and were referred to an appointment if the result remained ≥6 mIU/L (Figure 1).

Extreme preterm babies (children born at fewer than 32 weeks of gestation or weighing less than 1500 g), newborns with hemodynamic instability, and those who received a transfusion before sample collection for screening, who were at risk for a false-negative result, underwent serial collections even if the first sample was normal (special protocols). In such instances, bloodspots are repeated at 10 and 30 days of life or 10 days after a blood transfusion. Newborns are closely monitored, with guidance provided to the local medical team by the pediatric endocrinologists of the program through phone contact with NUPAD.

The first consultations are conducted by the PTN-MG team at the reference center (Hospital das Clínicas—UFMG, in Belo Horizonte). Subsequent follow-up, including quarterly assessments up to 3 years of age, is carried out collaboratively at both the reference center and in the patient’s municipality of residence.

Diagnosis is confirmed based on serum TSH (s-TSH—reference values (RV): 0.69 to 8.55 μIU/mL) and free thyroxine levels (fT4—RV: 0.89 to 1.76 ng/dL) by means of chemiluminescence immunometric assay (ICMA). Newborns with b-TSH between 6 and <10 mIU/L underwent thyroid ultrasound by one single trained radiologist, with a reference value for thyroid volume of 0.45 to 1.34 cm^3^ [18].

The criteria used to confirm cases and initiate treatment were s-TSH ≥ 10 µIU/mL and normal or low fT4 or borderline s-TSH (above the reference value, but lower than 10 µIU/mL) associated with low fT4. Patients diagnosed based on this last criterion were assessed for central CH.

Patients presenting with s-TSH ≥ 10 µIU/mL and fT4 below the reference value were classified as decompensated CH.

According to the sample size calculation, with 95% confidence, a minimum sample size was established for 60 detected cases of CH and at least 150,000 negative tests.

The analyses were performed using software R 4.1.2 and IBM SPSS Statistics 26. A significance level of 5% was considered for all the tests. Variables were verified for normality using the Kolmogorov–Smirnov or Shapiro–Wilk tests. Continuous variables were characterized by medians, minimum and maximum values, and percentiles. Comparisons between continuous variables were performed using the Kruskal–Wallis test with the Bonferroni post-hoc test. Categorical variables were represented by their absolute values and percentages and were compared using Pearson’s chi-square test. The incidence of CH was calculated, as well as the recall rates for new sample collections and calls for appointments in the period. The sensitivity and specificity of b-TSH cut-off values of 6, 7, 8, 9, and 10 mIU/L were used to create the ROC curve. Moreover, the respective positive and negative predictive values (PPVs and NPVs) were calculated to compare the performance. Patients who were screened using special protocols were excluded.

For clinical and laboratory comparisons, subjects referred for medical appointments during the study period were divided into two groups, stratified according to b-TSH values at first sampling: (1) newborns with b-TSH between 6 and <10 mIU/L and (2) newborns with b-TSH ≥ 10 mIU/L.

## 3. Results

During the study period, 162,729 newborns were screened in the PTN-MG, and 3070 by using special protocols. A total of 123 cases of CH were diagnosed and treated with levothyroxine, and 50 of them were identified due to a change in the b-TSH cut-off (Figure 2). The incidence for the cut-off of 6 mIU/L was 1 case for every 1323 newborns, whereas for the cut-off of 10 mIU/L, it was 1:2229 in the study period.

The recall rate was calculated for a b-TSH cutoff of 6 mIU/L. The recall rate for a second sample of neonatal screenings was 0.57%, and for medical appointments was 0.08%.

The area under the ROC curve for the first sample was 0.999, with a 95% confidence interval (95% CI) between 0.999 and 0.999; for the second sample, it was 0.991 (CI 0.986–0.996). The projected performances for the different cut-offs are summarized in Table 1. All the cut-offs assessed had adequate specificity and NPV. Lower cut-offs were related to better sensitivity, but with a decrease in PPV and increase in recall rates.

Group 1 consisted of 50 patients, with 27.6% preterm children (<37 weeks of gestation) and Group 2 had 58 patients, 20% premature. The prematurity rate did not differ significantly between the groups (*p* = 0.378). Newborns called for a medical appointment showed a median of b-TSH in the first neonatal screening sample of 7.44 mIU/L (Q1, 6.02; Q3, 9.91) in Group 1, and of 39.88 mIU/L (Q1, 10.90; Q3, 313) in Group 2 (*p* < 0.01). First samples were collected at a median age of 5 days for both groups. In the second sample, Group 1 had a b-TSH median of 9.53 mIU/L (Q1, 4.69; Q3, 24.34), at a median age of 14 days, and Group 2, at 13 days, had a median of 21.40 mIU/L (Q1, 10.68; Q3, 77.46) (*p* < 0.01). Patients with a b-TSH ≥ 20 mIU/L in the first sample were referred for emergency appointments.

In Group 1, 83% of patients showed at least one clinical sign of CH, such as prolonged jaundice, dry skin, poor weight gain, large fontanelles, and cranial sutures, and 94% in Group 2. Group 2 patients presented significantly more clinical signs than Group 1 (*p* = 0.02), even though they were assessed in a medical appointment earlier (median age of 16 vs. 25 days old).

Patients in Group 1 presented significantly lower levels of s-TSH and significantly higher levels of fT4 than those in Group 2 (*p* < 0.001 for both analyses), as summarized in Table 2. 21% of patients in Group 1 presented fT4 below the reference values used in the program.

A total of 18 patients from Group 1 underwent thyroid ultrasound upon diagnosis: three (16.7%) were suggestive of dysgenesis—two with hypoplasia, and one who showed hemiagenesis—and 15 (83.3%) were inconclusive—two with goiter and 13 with normal thyroid. Patients in Group 2 did not undergo ultrasound at the time of diagnosis. All patients in both groups will be re-evaluated at the age of 3 years with imaging tests after temporarily discontinuing medication.

## 4. Discussion

The decrease in the b-TSH cut-off from 10 to 6 mIU/L represented a significant increase in the test sensitivity, allowing the detection of more newborns with CH, but, also, a high rate of false-positive cases. The unequivocal benefits of treating children preemptively and reports of undiagnosed CH with current cut-offs of b-TSH have led many newborn screening programs to reduce the thresholds, resulting in a higher incidence of the disease [2,11,12,13].

The incidence of CH in Brazil ranges from 1:2500 to 1:4800 of live births with a b-TSH cut-off of 10 mIU/L [19]. Similar to what was found by the present study, the incidence of CH rises sharply to 1:950–1:1560 [20,21,22] in some states that reduced the cut-offs to 4.5 to 6 mIU/L.

The same have been reported in several parts of the world, with identification of mild CH cases and reduction in percentage of dysgenetic glands [4,11].

In accordance with these reports, we observed a greater proportion of patients with less severe conditions, with few or no symptoms and compatible laboratory findings. Most patients assessed did not present anatomical abnormalities on thyroid ultrasound, a frequent finding related to CH diagnoses with thyroid gland in situ after a reduction in the cut-off points [4,11,16,23,24]. In a survey conducted at the PTN-MG between 2018 and 2019, with 44 children under the unchanged protocol, the proportion of newborns with a thyroid with no anatomical alterations on ultrasound in the first appointment was 58%, which is lower than the current findings.

However, more severe cases of CH have been reported with lower cut-off points, and approximately 40% of patients with permanent and decompensated CH upon diagnosis were found with b-TSH between 8 and 10 mIU/L [25].

We observed although most newborns in Group 1 presented mild laboratory alterations, 21% of the sample presented decompensated CH, which is an unequivocal indication for treatment in all consensus on the disease [1,3,26].

We show that if the cut-off of 10 mIU/L had been kept, 40% of newborns would not have been diagnosed with the disease, which seems an unacceptable rate. A significant number of children with transient or permanent CH being left undiagnosed using the cut-off of 10 mIU/L (18%) led the authors suggesting a value of 6 mIU/L to be more appropriate for the screening program in the United Kingdom [13].

On the other hand, a problem related to the reduction in the b-TSH cut-off is a great number of children called for medical assessment.

Keeping higher cut-offs, as seen in some high income countries, can reduce the number of false-positive results [27]. However, active vigilance and quality infant care must be guaranteed to diagnose possible cases missed during screening. A Swedish study showed that using a b-TSH cut-off of 15 mIU/L, 50% of detected CH cases had come back negative during neonatal screening and were identified during pediatric care [28]. Taking care of these children at the right time is certainly not possible in many regions.

There is no consensus on the ideal recall rate. Just as for the cut-off value, this rate depends on several factors, such as the geographical coverage of the program, available infrastructure, and financial resources, among others. The worldwide rate of calls for medical appointments varies between 0.01% and 13.3%, depending on the adopted b-TSH cut-off [29].

We observed the reduction in the b-TSH cut-off was associated with a 5.2-fold increase in the rate of recall for a second sample, and as the result was normal for most children, the rate of calls for appointments increased by 30%. This is an undesirable consequence. Even if the second sample is collected close to the child’s residence and the local agents are trained to inform families about the need for recollection, there may be a negative impact. Furthermore, potential extra cost may be imposed to the screening program, and should be evaluated in the future.

A similar reduction, from 10 to 6 mIU/L in the United Kingdom, led the recall rate to increase from 0.08% to 0.23% [12]. The recall rate increased in all reported programs that implemented a reduction in the cut-off screening [30,31], up to 10 times [24,32]. High recall rates represent higher costs for screening programs, in addition to a higher number of healthy children undergoing needless exams and appointments, with a relevant emotional impact for the patients and their families.

In the United States, costs incurred from repeated tests can reach up to USD 2 million annually [29]; the burden can be expected to be bigger for disadvantaged populations. It is clear, however, that the lower the recall rate—without leaving sick children undiagnosed—the better.

In addition to the costs, a definition of the appropriate cut-off point for a neonatal screening program must consider the methods used; the median age of the newborns during sample collection; ethnicity of the population and degree of iodine sufficiency; the capacity to process the tests; the availability of specialized centers; and the socioeconomic and cultural factors. This definition should aim for a sensitivity close to 100%, but without overburdening the program with too many false-positive results.

Herein, the different cut-off points presented good specificity and a high NPV, which were expected based on the low prevalence of CH in the population [29]. Therefore, sensitivity and PPV assessments are more important for suitability analysis. Despite a higher sensitivity of the 6 mIU/L cut-off in comparison to the 10 mIU/L, the low PPV in the first sample—which generated over 88% of false-positive results—is not attractive to a government-funded program. In Australia, reducing the cut-off from 15 to 8 mIU/L caused the PPV to drop from 74.3% to 12.8% [32]. The same was reported in Brazilian regions—Rio de Janeiro and Campinas—which showed false-positive results close to 90% [20,33].

We sought better results to enhance the rate of CH diagnosis without significantly burdening the program or penalizing families. Analyzing the performances of the different cut-offs, we found an intermediate value of 8 mIU/L was related to good sensitivity and a PPV over two times higher than the cut-off of 6 mIU/L, which potentially makes it more adequate for the first sample. Thus, we opted to adopt different thresholds for the first and subsequent samples; 8 mIU/L for the first and 6 mIU/L for subsequent ones.

Although a higher prevalence of preterm infants could have been expected in Group 1, the similar number of preterm newborns observed in both groups precludes any effect of prematurity on the performance analysis of different b-TSH cut-off points, potentially increasing false-negative results.

One strategy to enhance the effectiveness of screening programs is re-screening extremely preterm and very low birth weight infants to prevent overlooking delayed TSH rise [1], and to reduce the recall rate for false-positive children with borderline results, without delaying the assessment of urgent cases [34].

Another one is the use of lower cut-off points only for the second sample, when recommended [34,35]. Decreasing the cut-off only in the second sample doubled the incidence of newborns diagnosed with dyshormonogenesis and dysgenesis [16], and allowed the diagnosis of 43.7% of CH cases in another report [36]. The use of these lower cut-off points takes into consideration the physiological changes of thyroid function, increase in TSH in the first 48 h after birth, and progressive decrease during the first weeks of life. This strategy appeared to be the best choice for improving our program, as the cut-off point of 6 mIU/L in the first sample resulted in an excessive number of false positives, although its use in subsequent samples increased test sensitivity without compromising PPV.

The disadvantages of using different cut-off points in the subsequent samples include the additional complexity of the screening process, which requires different criteria to interpret results according to the sample. This can increase the possibility of interpretation errors and demands more training of the healthcare professionals involved in the program. To address this issue, the PTN-MG has a supportive team that receives periodic training.

The main objective of CH screening is to prevent intellectual deficit associated with the disease. Therefore, the main argument of authors who are against reducing cut-off points is that the benefit of treatment for very mild and subclinical cases—which comprise the majority of cases that have been identified—is still under investigation. The results obtained in the treatment of moderate to severe CH since screenings were implemented should not be simply extrapolated [37,38]. Natural evolution studies have been conducted to show the developmental impacts suffered by children with untreated mild CH, but results are still conflicting. Some reports suggest differences in the neurodevelopment of children with borderline b-TSH were not clinically relevant to recommend changes in the cut-off [39,40,41], while others show that these children had an increased risk of poor school performance, developmental changes, reduced memory span, and a higher chance to develop special needs [42,43]. This assessment is complex since cognitive deficits can be associated to multiple factors, including socioeconomic and environmental factors [44,45].

On the other hand, excessive treatment in CH patients appears to be linked to unfavorable cognitive outcomes [46,47,48]. There is still no consensus on this matter; further research is needed.

The main limitation of this study is a potential underestimation of false-negative cases since all patients with a neonatal b-TSH < 6 mIU/L were considered normal. Furthermore, few patients underwent thyroid ultrasound. However, the sample size and rigorous protocol are strengths supporting the results found are highly consistent.

## 5. Conclusions

There was an increase in the diagnosis of CH after reducing the b-TSH cut-off to 6 mIU/L, mainly attributed to the detection of children with mild cases. Maintaining a cut-off of 10 mIU/L would have resulted in 40% of newborns with CH going undiagnosed, which appears to be an unacceptable rate. Since the b-TSH cut-off of 6 mIU/L for the first sample exhibited a very low PPV, an intermediate value of 8 mIU/L is proposed, along with a threshold of 6 mIU/L for subsequent samples, to achieve the expected results. Continuous longitudinal monitoring of newborn screening programs utilizing lower cut-off points is essential to evaluate their efficacy over time.

## Figures and Tables

**Figure 1 IJNS-10-00052-f001:**
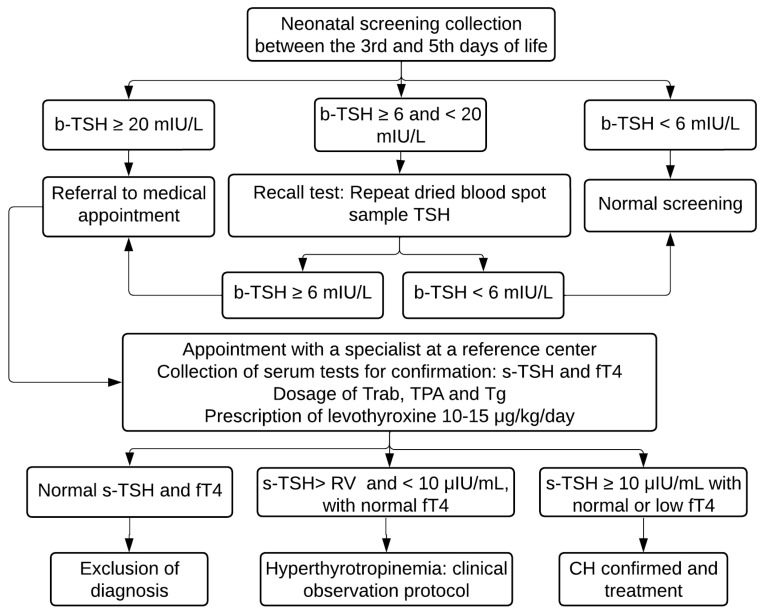
PTN-MG flowchart for diagnosis of congenital hypothyroidism between November 2021 and August 2022. b-TSH: filter-paper blood-spot TSH; s-TSH: serum TSH; fT4: free T4; Trab: TSH receptor autoantibodies; TPA: Thyroid Peroxidase Antibody; Tg: Thyroglobulin; RV: reference value.

**Figure 2 IJNS-10-00052-f002:**
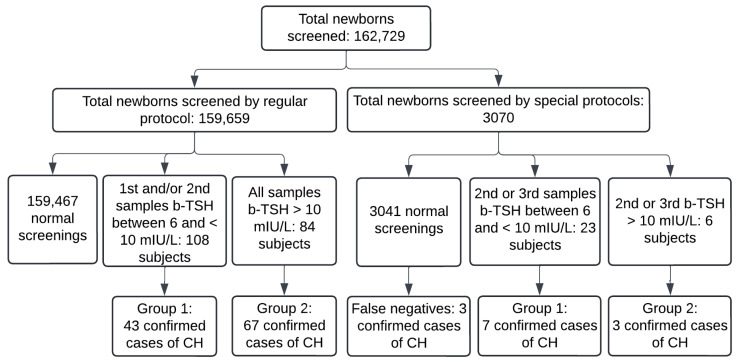
Flowchart of cases of congenital hypothyroidism diagnosed between November 2021 and August 2022 in Minas Gerais. b-TSH: filter-paper blood-spot TSH; CH: congenital hypothyroidism. Normal screenings include subjects who had a normal first sample and those whose screening was normalized in the second sample. False negatives: three confirmed cases diagnosed by clinical signs in the period.

**Table 1 IJNS-10-00052-t001:** Projected performance for different b-TSH cut-offs for congenital hypothyroidism in neonatal screening between November 2021 and August 2022, in Minas Gerais State.

b-TSH Cut-Off (mIU/L) for the 1st Sample (*n* = 159,659)	Sensitivity (95% CI)	Specificity (95% CI)	PPV (95% CI)	NPV (95% CI)	Recall Rate for 2nd Sample
6	100% (100–100)	99.50% (99.46–99.53)	11.87% (11.71–12.03)	100.00% (100–100)	0.57%
7	85.30% (85.15–85.49)	99.70% (99.67–99.73)	16.88% (16.69–17.06)	99.99% (99.99–99.99)	0.35%
8	78.90% (78.7–79.1)	99.80% (99.82–99.86)	24.93% (24.72–25.14)	99.99% (99.98–99.99)	0.22%
9	70.60% (70.42–70.87)	99.90% (99.89–99.92)	34.69% (34.45–34.92)	99.98% (99.97–99.99)	0.14%
10	64.20% (63.99–64.46)	99.90% (99.93–99.95)	41.42% (41.18–41.66)	99.98% (99.97–99.98)	0.11%
**b-TSH cut-off (mIU/L)** **for the 2nd sample** **(*n* = 793)**	**Sensitivity** **(95% CI)**	**Specificity** **(95% CI)**	**PPV** **(95% CI)**	**NPV** **(95% CI)**	**Recall Rate for Appointment**
6	98.20% (97.33–99.16)	98.10% (97.15–99.05)	80.00% (77.22–82.78)	99.86% (99.60–100)	0.08%
7	80.70% (77.95–83.45)	98.10% (97.15–99.05)	76.67% (73.72–79.61)	98.50% (97.65–99.35)	0.07%
8	73.70% (68.80–75.06)	98.40% (97.49–99.25)	77.36% (74.45–80.27)	97.84% (96.83–98.85)	0.07%
9	64.90% (61.59–68.23)	98.90% (98.19–99.63)	82.22% (79.56–84.88)	97.33% (96.20–98.45)	0.06%
10	59.60% (56.23–63.06)	99.20% (98.56–99.81)	85.00% (82.51–87.49)	96.95% (95.75–98.14)	0.06%

b-TSH: filter-paper blood-spot TSH; CI: confidence interval; PPV: positive predictive value; NPV: negative predictive value.

**Table 2 IJNS-10-00052-t002:** Serum tests of congenital hypothyroidism newborns between 2021 and 2022 in Minas Gerais State.

	Serum TSH		Free T4	
	Group 1 (b-TSH 6–9.9 mIU/L)	Group 2 (b-TSH >= 10 mIU/L)	Group 1 (b-TSH 6–9.9 mIU/L)	Group 2 (b-TSH >= 10 mIU/L)
Median	14.52 *	177.7 *	1.14 *	0.56 *
Minimum	10.08	10.4	0.40	0.10
Maximum	193.65	1370.0	2.54	1.68
25th percentile	11.72	47.81	0.93	0.27
75th percentile	27.62	394.55	1.37	0.99

b-TSH: filter-paper blood-spot TSH. Serum TSH: reference value = 0.69 to 8.55 μIU/mL; Free T4: reference value = 0.89 to 1.76 ng/dL. Group 1: *n* = 50; Group 2: *n* = 58. * *p* < 0.001—Kruskal–Wallis test.

## Data Availability

Data are not available due to ethical reasons. Further enquiries can be directed to the corresponding author.

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
