# Peer review of "Impact of Lowering TSH Cut-Off on Neonatal Screening for Congenital Hypothyroidism in Minas Gerais, Brazil"

_2409-515X, 2024, doi:10.3390/ijns10030052_

Round 1

Reviewer 1 Report

Comments and Suggestions for Authors

This manuscript by Braga, N.T.P., et al. on the effect of lowering the TSH cutoff value in newborn screening for congenital hypothyroidism is significant and interesting because it examines sensitivity and positive predictive value in detail. However, just two points need to be confirmed.

My comments are as follows.

1. P1L28-30: “A proposed strategy is to adopt a higher b-TSH cut-off in the first sample and a lower one in the subsequent samples from the same child, which yields a greater number of diagnoses with an acceptable PPV.”

>>> The important suggestion that it is useful to adopt a higher TSH cutoff in the first sample and a lower cutoff in the second is not explicitly discussed in the text.

2. Table 2

>>> The maximum TSH value for Group 2 is listed as 1.370. It is confusing whether the "period" indicates a "digit separator" or a "decimal point."

Author Response

  1. P1 L28-30: “A proposed strategy is to adopt a higher b-TSH cut-off in the first sample and a lower one in the subsequent samples from the same child, which yields a greater number of diagnoses with an acceptable PPV.”

>>> The important suggestion that it is useful to adopt a higher TSH cutoff in the first sample and a lower cutoff in the second is not explicitly discussed in the text.

 Authors responseWe agree and have included a new paragraph in the discussion to enhance the clarity of the text (page 8, lines 294-297).

  1. Table 2

>>> The maximum TSH value for Group 2 is listed as 1.370. It is confusing whether the "period" indicates a "digit separator" or a "decimal point."

 Authors responseWe apologize for the mistake; change was made in Table 2 (page 6, line 180).

Reviewer 2 Report

Comments and Suggestions for Authors

This paper (IJNS-2987169, Nathalia Teixeira Palla Braga et al.) evaluates the screening efficiency, such as the recollection rate and positive predictive value, by changing the cutoff value of TSH for newborn screening in congenital hypothyroidism. The paper employed a large cohort, potentially providing some insights into the appropriate cutoff value. Despite some interesting points, various problems currently prevent this paper from being published in IJNS.

#1 It is predicable that lowering the cut off criteria of TSH would cause lower PPV and high recall rate for the 2nd blood sampling. 

Which point is the most novel and important aspect of this study?

#2 It is hard to catch the study design employed in the paper. 

The authors are encouraged to provide a diagram of this study to explain the structure of the study, including the number of subjects and patients in each category and group. It would facilitate the readers' understanding of the study. 

#3 There is a lack of uniformity in the use of terminology, which makes it difficult to understand the manuscript. 

For example,

L115

For clinical and laboratory comparisons, “patients” referred for medical appointments during the study period were divided into two groups, stratified according to b-TSH values at first sampling: 1) newborns with b-TSH between 6 and <10 mIU/L and 2) newborns with b-TSH ≥10 mIU/L.

Using “patients” is not appropriate because the newborns were not diagnosed to have hypothyroidism. “Newborns” or “subjects” should be used. 

L120: 162,730 patients: “patients” means those who were diagnosed hypothyroidism and treated with LT4? If not, “newborns” or “subjects” should be used. This paper is about newborn screening, and a term “patients” should be used for the subjects that were confirmed to have hypothyroidism. 

L137: “Group 1 consisted of 50 newborns” 

Do newborns mean false positive cases? If 50 cases have hypothyroidism, the authors should clarify it. 

#4

L120

“During the study period, 162,730 patients were screened using the PTN-MG, and 120

3,070 newborns.”

This may mean that the study population would be 159,660. 

But, the number indicated in Table 1 is 159,659. One case is missing. 

#5 It is noteworthy that among Group 2 patients, only those whose first b-TSH sample was between 10 and 20 mIU/L had a second sample evaluated.

I can not understand the sentence. 

Newborns whose TSH >20 would be referred to a hospital. The criteria for Group2 is TSH>10. So, it is quite natural that newborns whose first b-TSH sample was between 10 and 20 mIU/L had a second sample evaluatednot noteworthy. 

#6 If the authors would like to evaluate the efficiency of the screening, financial issues should be included in the study. The potential extra cost of the screening by taking a lower TSH criteria should be included in the study and precisely discussed. 

Author Response

This paper (IJNS-2987169, Nathalia Teixeira Palla Braga et al.) evaluates the screening efficiency, such as the recollection rate and positive predictive value, by changing the cutoff value of TSH for newborn screening in congenital hypothyroidism. The paper employed a large cohort, potentially providing some insights into the appropriate cutoff value. Despite some interesting points, various problems currently prevent this paper from being published in IJNS.

#1 It is predicable that lowering the cut off criteria of TSH would cause lower PPV and high recall rate for the 2nd blood sampling. 

Which point is the most novel and important aspect of this study?

Authors response: Indeed, it is predictable that reducing the TSH cutoff point in neonatal screening would result in a lower PPV and an increase in the recall rate. However, there is no consensus on the tresholds, and each program proposes a different cutoff. The main objective of the study was to determine the most appropriate cutoffs for the population of Minas Gerais, shedding light on potential alterations in other programs as well. We were able to conclude that a TSH cutoff point of 8mIU/L in the first sample and 6mIU/L in subsequent samples, would enhance the sensitivity of our public screening program, while maintaining acceptable levels of PPV and recall rate.

#2 It is hard to catch the study design employed in the paper. 

The authors are encouraged to provide a diagram of this study to explain the structure of the study, including the number of subjects and patients in each category and group. It would facilitate the readers' understanding of the study.

Authors responseWe agree and included a new diagram to enhance the clarity of the text (page 4, line 138, Fig 2).  #3 There is a lack of uniformity in the use of terminology, which makes it difficult to understand the manuscript. 

For example,

L115

For clinical and laboratory comparisons, “patients” referred for medical appointments during the study period were divided into two groups, stratified according to b-TSH values at first sampling: 1) newborns with b-TSH between 6 and <10 mIU/L and 2) newborns with b-TSH ≥10 mIU/L.

Using “patients” is not appropriate because the newborns were not diagnosed to have hypothyroidism. “Newborns” or “subjects” should be used. 

L120: 162,730 patients: “patients” means those who were diagnosed hypothyroidism and treated with LT4? If not, “newborns” or “subjects” should be used. This paper is about newborn screening, and a term “patients” should be used for the subjects that were confirmed to have hypothyroidism. 

L137: “Group 1 consisted of 50 newborns” 

Do newborns mean false positive cases? If 50 cases have hypothyroidism, the authors should clarify it.

Authors response: We agree and made changes marked-up in the manuscript. We kept the term “patients” only for those with confirmed congenital hypothyroidism (page 4, lines 127 and 132; page 5, lines 158, 159, 168 and 173).

#4

L120

“During the study period, 162,730 patients were screened using the PTN-MG, and 120

3,070 newborns.”

This may mean that the study population would be 159,660. 

But, the number indicated in Table 1 is 159,659. One case is missing.

Authors response: We apologize for the mistake; change was made to the manuscript (page 4, line 132).

#5 It is noteworthy that among Group 2 patients, only those whose first b-TSH sample was between 10 and 20 mIU/L had a second sample evaluated.

I can not understand the sentence. 

Newborns whose TSH >20 would be referred to a hospital. The criteria for Group2 is TSH>10. So, it is quite natural that newborns whose first b-TSH sample was between 10 and 20 mIU/L had a second sample evaluated, not noteworthy.

Authors response: We agree and removed the sentence, as it is redundant and doesn’t contribute to the understanding of the text.  

 #6 If the authors would like to evaluate the efficiency of the screening, financial issues should be included in the study. The potential extra cost of the screening by taking a lower TSH criteria should be included in the study and precisely discussed.

Authors responseWe agree, and are planning to evaluate extra cost of the screening as soon as we have a greater number of exams to allow robust results. We are aware of the benefits of identifying newborns which would not have the diagnosis of hypothyroidism till now. We included a sentence discussing this aspect (page 7, line 239-243).

Reviewer 3 Report

Comments and Suggestions for Authors

This is a paper nicely demonstrating how lowering the whole blood TSH levels in newborn screening affects important screening measures such as specificity and the positive predictive value for the screening program. The paper merits publication but can be slightly improved. Some suggestions and thoughts follow below.

Line 20. Not sure cost-benefit is the best expression to use since this paper does not take actual costs into account. Would suggest considering changing the phrasing (also later in the paper such as line 49).

Method section. What is the average transportation time from sampling to arrival in the laboratory? By postal service? What method (commercially available kit) and instrumentation(s) were used? Are results based on an average of two or three punches? Work Monday-Friday in the laboratory? What are the overall turnaround times in the screening laboratory? Are referrals made by telephone? Are there many (pediatric) reference centers receiving these referrals?

Line 85-88. What is meant by extreme preterm babies (in birth weight or gestational age)? Also, nice to define premature in line 137. Could be interesting to explain the special protocol used for preemies but since these babies are excluded in the study it may be reasonable to exclude this information.

Result and table 1. Is it possible for the authors to calculate/write average times for the first and second sampling and relate this to the information on line 149 when the children were seen by a physician?

How many of the children in group 2 had “clinical signs”?

Results such as in table 2 for the CH tests are given with many value digits probably not valid for the method used. Two or three value digits would be more appropriate and could be considered if possible.

Sampling a second time also increase the stress for families. Where is the second dried blood spot test taken and who explain the reason for this second sampling to the families? A short sentence on this could perhaps also be introduced in the discussion.

Would be interesting to follow these two groups to see how many children in each group that can stop hormone treatment at the age of three years.

Comments on the Quality of English Language

No more specific comments than suggested above.

Author Response

This is a paper nicely demonstrating how lowering the whole blood TSH levels in newborn screening affects important screening measures such as specificity and the positive predictive value for the screening program. The paper merits publication but can be slightly improved. Some suggestions and thoughts follow below.

Line 20. Not sure cost-benefit is the best expression to use since this paper does not take actual costs into account. Would suggest considering changing the phrasing (also later in the paper such as line 49).

Authors responseWe agree, and are planning to evaluate extra cost of the screening as soon as we have a greater number of exams to allow robust results. We included a sentence discussing this aspect (page 7, line 248-252), and rephrasing to “impact” (page 1, line 20 and page 2, line 49).

Method section. What is the average transportation time from sampling to arrival in the laboratory? By postal service? What method (commercially available kit) and instrumentation(s) were used? Are results based on an average of two or three punches? Work Monday-Friday in the laboratory? What are the overall turnaround times in the screening laboratory? Are referrals made by telephone? Are there many (pediatric) reference centers receiving these referrals?

Authors responseWe included a new paragraph answering these questions. (page 2, lines 72 to 84).

Line 85-88. What is meant by extreme preterm babies (in birth weight or gestational age)? Also, nice to define premature in line 137. Could be interesting to explain the special protocol used for preemies but since these babies are excluded in the study it may be reasonable to exclude this information.

Authors responseWe included the meaning of extreme preterm babies (page 3, lines 89, 90 and page 5, line 157), and a brief summary of the special protocols (page 3, lines 94 to 96).

Result and table 1. Is it possible for the authors to calculate/write average times for the first and second sampling and relate this to the information on line 149 when the children were seen by a physician?

Authors responseThis information has been added to the text. (page 5, line 162-164 and 172).

How many of the children in group 2 had “clinical signs”?

Authors responseThe percentage of patients with clinical signs in each group was added to the text. (page 5, lines 168-169).

Results such as in table 2 for the CH tests are given with many value digits probably not valid for the method used. Two or three value digits would be more appropriate and could be considered if possible.

Authors responseWe apologize for the mistake; change was made in Table 2 (page 6, line 179).

Sampling a second time also increase the stress for families. Where is the second dried blood spot test taken and who explain the reason for this second sampling to the families? A short sentence on this could perhaps also be introduced in the discussion.

Authors responseWe agree, and a paragraph was included in the discussion (page 7, line 239 to 243).

Would be interesting to follow these two groups to see how many children in each group that can stop hormone treatment at the age of three years.

 Authors responseThe evaluation of newborns undergoing hypothyroidism treatment at three years old is already included in the follow-up protocol (imaging tests - ultrasound and scintigraph, after temporarily stopping the medication). We are planning to re-evaluate all patients and report the results.

Reviewer 4 Report

Comments and Suggestions for Authors

The manuscript "Impact of lowering TSH cut-off on neonatal screening for con-2 genital hypothyroidism in Minas Gerais, Brazil" give interesting results on the impact of lowering TSH cut-off level for CH screening.

The paper is well wrtitten and just need some improvement before publication.

The results highlight that until 40% of newborns affected with CH may be missed with cut-off above 10. The authors may explain wether these newborns were finally treated for CH of if it's just transient CH. If these babies are mainly transient forms authors should explain the benefit of their screening.

Some minor points :

- P2 / L61 : please give the reference for the ethic comitee

- P2 / L75 : precise the name of the TSH kit provider

- P2 / L79 : what is the temporality for the second sample

-P5 / L161 : which type of diagnosis are made in the group 2 : how many dysgenesis, ....

Author Response

The manuscript "Impact of lowering TSH cut-off on neonatal screening for con-2 genital hypothyroidism in Minas Gerais, Brazil" give interesting results on the impact of lowering TSH cut-off level for CH screening.

The paper is well wrtitten and just need some improvement before publication.

The results highlight that until 40% of newborns affected with CH may be missed with cut-off above 10. The authors may explain wether these newborns were finally treated for CH of if it's just transient CH. If these babies are mainly transient forms authors should explain the benefit of their screening.

 Authors responseAs we have discussed in the paper, programs that have reduced the thresholds of b-TSH have identified a a significant percentage of mild and transitory CH cases, alongside permanent and decompensated cases. The median age at which our patients presented for medical evaluation was 16 days in group 2 and 23 days in group 1. We considered it a risk not to initiate treatment for these patients, mainly considering that even those with transient hypothyroidism may benefit from treatment. Therefore, all newborns whose serum TSH test was above 10 had a confirmed diagnosis and treatment was initiated (this information has been added on page 4, line 133 to 134). However, we acknowledge that some patients, particularly in Group 1,may have a transient course  and could discontinue medication. These patients are being monitored. At the age of three years, all patients in both groups will undergo thyroid ultrasound and scintigraphy, in addition to serum tests, to determine the etiology and assess the possibility of transient hypothyroidism (after discontinuing treatment for one month). According to our protocol, only patients who maintain suppressed TSH and elevated T4 levels using the lowest commercial dose available (12.5 mcg of levothyroxine) can discontinue medication before the age of three. As the studied patients are currently between one and two years old, we don’t yet have a definitive answer to this question.

Some minor points :

- P2 / L61 : please give the reference for the ethic comitee

Authors responseThe approval number from the ethics committee was included. (page 2, line 61).

- P2 / L75 : precise the name of the TSH kit provider

Authors responseThe name of the TSH kit provider was added. (page 2, line 79 and 80).

- P2 / L79 : what is the temporality for the second sample

Authors responseThis information has been added to the text. (page 2, line 83; page 5, line 164 and 165).

-P5 / L161 : which type of diagnosis are made in the group 2 : how many dysgenesis, ....

 Authors responseAs ultrasound exams were not conducted on patients from Group 2, we are unable to ascertain their hypothyroidism etiologies. The evaluation of newborns undergoing hypothyroidism treatment at three years old is already included in the follow-up protocol (imaging tests - ultrasound and scintigraph, after temporarily stopping the medication). Thus, we will be able to accurately identify the etiology of each patient's hypothyroidism, as well as determine the percentage of transient cases in each group (page 6, 188 to 190).

Round 2

Reviewer 2 Report

Comments and Suggestions for Authors

The authors corrected the manuscript appropriately.

Author Response

Ms. Katarina Bojkovic
Assistant Editor, MDPI Belgrade

We changed the manuscript “Impact of reducing the TSH cutoff point in neonatal screening for congenital hypothyroidism in Minas Gerais, Brazil” according to the reviewer's comments. We would like to thank you and the reviewer for your suggestions. Please see responses to comments below.

Kind regards,

Dr. Nathalia Teixeira Palla Braga.

Reviewer 2

1- Line 123 – “Using the ROC curve, the sensitivity and specificity of b-TSH cut-off values of 6, 7, 8, 9, and 10 mIU/L were calculated” is backwards.  It should be “The sensitivity and specificity of b-TSH cut-off values of 6, 7, 8, 9, and 10 mIU/L were used to create the ROC curve.”

Authors responseWe agree and made changes marked-up in the manuscript (lines 122-123).

2- The discussion is a little long.  Lines 279-282 and 317-321 do not seem to be relevant to this manuscript.

Authors responseWe agreed and simplified the paragraphs in the manuscript (lines 274-277 and 307-309).

Minor points:

3- In the Abstract, “lower serum TSH (p<0.01),” should be removed because the serum TSH must be lower when the group is defined by the lower TSH.

Authors responseIndeed, it is expected that reducing the TSH cutoff in neonatal screening would result in lower serum TSH levels in the neonates. However, since we also observed some significantly elevated serum TSH results in this group of infants, we opted to include this information in the abstract.

4- Line 65 - “reference value” should be “cut-off value”

Authors responseWe agree and made changes marked-up in the manuscript (line 65).

5- Figs. 1 and 2 are blurry.  A better screenshot from the original source is needed.

Authors responseWe agree and made changes in the figures 1 (line 85) and 2 (line 136).

6- Line 112 – “sample calculation” should be “sample size ,calculation”, but I am not aware of any sample size calculation that uses “accuracy” in the calculation.  Please correct or provide a citation.

Authors responseWe corrected this paragraph in the manuscript (lines 112-113).

7- Line 145 – “The recall rate for the second sample of neonatal screenings was 0.57%” would be clearer if stated as “The recall rate for getting a second sample of neonatal screenings was 0.57%.”  The second sample only has a "recall rate for appointment".

Authors responseWe agree and made changes marked-up in the manuscript (lines 142-144).

8- Line 194 – “precociously” should be “preemptively”; The former means “too early”

Authors responseWe agree and made changes marked-up in the manuscript (lines 190-191).

9- Line 195 – “been” needs to be removed

Authors responseWe removed this word in the manuscript.

10- line 200 – “drop” should be “rises”;  The incidence is higher even though the number is smaller.

Authors responseWe apologize for the mistake; change was made in the manuscript (line 196)

11- Line 202 – I don’t understand, “paralleled by a change in newborns profile.”

Authors responseWe removed this part of the sentence as it was not relevant to understanding the data provided in the paragraph (lines 198-199).